# Association of Low Skeletal Muscle Mass with the Phenotype of Lean Non-Alcoholic Fatty Liver Disease

**DOI:** 10.3390/healthcare10050850

**Published:** 2022-05-05

**Authors:** Jun-Hyeon Byeon, Min-Kyu Kang, Min-Cheol Kim

**Affiliations:** Department of Internal Medicine, College of Medicine, Yeungnam University, Daegu 42415, Korea; junhyun2010@naver.com (J.-H.B.); kimmc1358@naver.com (M.-C.K.)

**Keywords:** non-alcoholic fatty liver disease, skeletal muscle, lean NAFLD, body mass index

## Abstract

Low skeletal muscle mass (LSMM) has emerged as a potential risk factor for non-alcoholic fatty liver disease (NAFLD). However, its clinical implications in patients with lean NAFLD have not yet been elucidated. We investigated the role of LSMM in patients with lean NAFLD. Lean NAFLD was defined as a body mass index of ≤23 kg/m^2^. Using bioelectrical impedance analysis, the appendicular skeletal muscle mass (ASM) was adjusted by height squared. The LSMM was based on 1 SD below the sex-specific mean for young, healthy Asian adults. Of the 8905 ultrasound-confirmed NAFLD patients, 3670 (41.2%) were diagnosed with lean NAFLD. The lean NAFLD group was younger (45.0 vs. 49.0 years, *p* < 0.001), and had a lower waist circumference (74.0 vs. 85.0 cm, *p* < 0.001), lower prevalence of diabetes (3.1 vs. 7.4%, *p* < 0.001) and hypertension (4.2 vs. 15.2%, *p* < 0.001), and a higher proportion of LSMM (28.0 vs. 2.2%, *p* < 0.001) than the non-lean NAFLD group. Stepwise adjusted models suggested that LSMM was associated with lean individuals with NAFLD (odds ratio = 7.02, *p* < 0.001). LSMM may be a novel risk factor for lean NAFLD patients more than non-lean NAFLD patients, independent of classic metabolic factors.

## 1. Introduction

Non-alcoholic fatty liver disease (NAFLD), characterized by >5% hepatic steatosis without secondary causes, including excessive alcohol consumption, is strongly associated with type 2 diabetes mellitus (T2DM), obesity, and insulin resistance (IR) [1,2]. The prevalence of NAFLD is approximately 25–40% of the global population, and is expected to increase in accordance with increasing metabolic derangements [2,3].

Although NAFLD mostly occurs in obese patients, it can also occur in lean/nonobese individuals [4]. Based on body mass index (BMI), lean NAFLD is defined as NAFLD with a BMI < 23 kg/m^2^ in the Asian population and BMI < 25 kg/m^2^ in the non-Asian population [5]. The main pathophysiology and natural course of lean/nonobese NAFLD are not yet fully understood. In some studies, sedentary lifestyle, visceral fat accumulation, and genetic factors, such as the rs738409 variant of the patatin-like phospholipase 3 (PNPLA3) gene, are closely associated with lean/nonobese NAFLD [6,7]. According to recent studies, lean NAFLD has more favorable metabolic profiles than non-lean NAFLD. In contrast, lean NAFLD has a higher risk for the development of liver-related events than non-lean NAFLD and is associated with all-cause and cardiovascular disease mortality independent of metabolic risk factors [8,9,10,11].

Sarcopenia, a decrease in skeletal muscle mass and strength with age, may be a risk factor for NAFLD and significant or advanced fibrosis in patients with NAFLD [12,13,14]. However, studies demonstrating an association between low skeletal muscle mass (LSMM) and lean NAFLD are not well known. Considering that the skeletal muscle is a metabolic organ and is associated with IR and chronic inflammation, the role of skeletal muscle mass as a potential risk factor needs to be elucidated in patients with lean NAFLD. We aimed to investigate the association between LSMM and lean and non-lean NAFLD.

## 2. Materials and Methods

### 2.1. Study Participants and Design

We retrospectively reviewed the anthropometric and clinical data of 41,313 Korean adults (aged ≥20 years) who underwent health screening at a health promotion center of Yeungnam University Hospital in South Korea between January 2013 and December 2019. A total of 32,408 participants were excluded based on the following criteria: (i) participants without definite evidence of fatty liver on abdominal ultrasound (*n* = 29,861); (ii) participants with positive serological results for hepatitis B (*n* = 517) or C (*n* = 18); (iii) participants who had a history of excessive alcohol consumption (men > 140 g/week and women > 70 g/week) (*n* = 521) [7]; and (iv) participants with inadequate data (*n* = 1491). Finally, 8905 patients with NAFLD were included in the study (Figure 1).

The requirement for informed consent was waived owing to the retrospective study design. The study protocol was approved by the institutional review board of Yeungnam University Hospital (YUMC-2021-05-031).

### 2.2. Assessment of Anthropometric, Clinical, and Laboratory Data

During the health screening examination, anthropometric variables including height, weight, waist circumference (WC), and seated blood pressure (BP) were measured by trained nurses. Data on alcohol consumption, personal medical history, and medication use were obtained from a self-administered questionnaire. Blood samples and hepatobiliary ultrasound (US) findings were collected in the morning after at least 8 h of overnight fasting. The blood sample consisted of a complete blood count and liver, glucose, and lipid profiles.

Diabetes mellitus (DM) was defined as a fasting plasma glucose (FPG) ≥ 126 mg/dL, HbA1c level ≥6.5%, or a history of consuming antidiabetic medication [15]. Hypertension was defined as a systolic BP ≥ 140 mmHg, diastolic BP ≥ 90 mmHg, or a history of antihypertensive medication use. Based on the International Diabetes Federation criteria, metabolic syndrome in Asian adults was defined as visceral obesity (WC ≥ 90 cm in men and ≥85 cm in women) plus any two of the following factors: increased levels of triglycerides (≥150 mg/dL), FPG (≥100 mg/dL), elevated BP (systolic BP ≥ 130 mmHg or diastolic BP ≥ 85 mmHg), and decreased high-density lipoprotein cholesterol (≤40 mg/dL in men and ≤50 mg/dL in women) [16]. The presence of IR was defined as HOMA-IR ≥ 2.5 [17].

### 2.3. Assessment of NAFLD and Lean NAFLD

Using EPIQ 5 and EPIG 6 (Koninklijke Philips NV, Amsterdam, The Netherlands), our previous publications outlined the definition of fatty liver and NAFLD adopted by the Asia–Pacific guidelines [7,12]. Lean NAFLD was defined as a BMI ≤ 23 kg/m^2^ in patients with NAFLD [18].

### 2.4. Assessment of LSMM

Bioelectrical impedance analysis using InBody 720 (Biospace, Seoul, Korea) was performed by trained staff on the same day. The appendicular skeletal muscle mass (ASM) was defined as the sum of the muscle mass of the four limbs. Skeletal muscle index was defined as ASM divided by height squared (ASM/ht^2^). LSMM was defined as ASM/ht^2^ using sex-different cutoff values of <7.0 kg/m^2^ for men and <5.8 kg/m^2^ for women, according to the Consensus Report of the Asian Working Group for Sarcopenia [19,20]. In our study, the term sarcopenia was not used because of the absence of muscle function tests, including the grip power test.

### 2.5. Statistical Analysis

Continuous variables are expressed as mean ± standard deviation using Student’s t-test, and categorical variables are expressed as *n* (%) using the chi-squared test. The association between the presence of lean NAFLD and LSMM in patients with NAFLD was assessed using logistic regression analysis. Sequential adjusted models applied to the traditional confounding factors for NAFLD were analyzed. Model 1 was adjusted for age and gender. Model 2 was adjusted for DM, hypertension, and WC, as well as Model 1 parameters. Model 3 was adjusted for the presence of IR and C-reactive protein, as well as Model 2 parameters. Model 4 was adjusted for serum alanine aminotransferase, gamma-glutamyl transferase, triglyceride levels, and platelet counts, as well as Model 3 parameters. All analyses were performed using the R software (version 3.0.2; R Foundation for Statistical Computing, Vienna, Austria), and statistical significance was set at *p* < 0.05.

## 3. Results

### 3.1. Baseline Characteristics

Table 1 shows the baseline characteristics of patients with NAFLD according to the presence or absence of lean NAFLD. Among the 8905 NAFLD patients, 3670 (41.2%) were lean NAFLD patients. Lean NAFLD patients were younger (45.8 vs. 49.0 years, *p* < 0.001), more likely to be female (66.6% vs. 34.2%, *p* < 0.001), had a lower BMI (20.9 vs. 25.9 kg/m^2^, *p* < 0.001) and WC (73.5 vs. 85.8 cm, *p* < 0.001), and had a lower prevalence of DM (3.1% vs. 7.4%, *p* < 0.001), hypertension (4.2% vs. 15.2%, *p* < 0.001), and metabolic syndrome (3.7% vs. 17.3%, *p* < 0.001) than non-lean NAFLD patients. The lean NAFLD group had lower levels of liver and metabolic profiles, including alanine aminotransferase, platelet, gamma-glutamyl transferase, lipid profiles, and HOMA-IR, than the non-lean NAFLD group.

Moreover, the lean NAFLD group had a lower skeletal muscle mass index using the Student’s t-test (6.2 vs. 7.8, kg/m^2^, *p* < 0.001) and a higher percentage of LSMM (28.0% vs. 2.2%, *p* < 0.001) than the non-lean NAFLD group using the chi-squared test. In the sex-specific subgroup analyses, skeletal muscle mass was lesser in the lean men-NAFLD group (7.4 vs. 8.2 kg/m^2^, *p* < 0.001) and women-NAFLD group (6.0 vs. 6.6 kg/m^2^, *p* < 0.001) than in the non-lean sex-specific NAFLD subgroup using the Student’s t-test. In addition, the percentage of LSMM was lower in the lean men-NAFLD group (1.5% vs. 24.3%, *p* < 0.001) and women-NAFLD group (3.6% vs. 29.8%, *p* < 0.001) than in the non-lean sex-specific NAFLD group using the chi-squared test (Figure 2).

### 3.2. Association between Lean NAFLD Individuals and LSMM in Patients with NAFLD

In multivariate analysis, age (odds ratio (OR), 0.99; 95% CI, 0.98–0.99; *p* < 0.001), male sex (OR, 0.51; 95% CI, 0.44–0.60; *p* < 0.001), WC (OR, 0.67; 95% CI, 0.66–0.68; *p* < 0.001), hypertension (OR, 0.52; 95% CI, 0.40–0.68; *p* < 0.001), presence of IR (OR, 0.68; 95% CI, 0.53–0.87; *p* = 0.003), triglyceride level (OR, 0.86; 95% CI, 0.77–0.95; *p* = 0.005), platelet count (OR, 0.82; 95% CI, 0.72–0.93; *p* = 0.004), and LSMM (OR, 6.88; 95% CI, 5.27–9.07; *p* < 0.001) were significant risk factors for lean NAFLD patients (Table 2).

To evaluate the role of LSMM in patients with lean NAFLD, we performed multivariate analysis using adjusted models. Table 3 shows the adjusted OR of the LSMM for individuals with lean NAFLD. The presence of LSMM was a significant independent risk factor for lean NAFLD individuals, which was maintained after stepwise adjustment for age and sex (Model 1: OR, 18.86; 95% CI, 15.34–23.40; *p* < 0.001); for DM, hypertension, and WC (Model 2: OR, 6.79; 95% CI, 5.20–8.97; *p* < 0.001); for the presence of IR and CRP (Model 3: OR, 6.88; 95% CI, 5.26–9.09; *p* < 0.001); and for serum alanine aminotransferase, gamma-glutamyl transferase, triglyceride levels, and platelet count (Model 4: OR, 7.02; 95% CI, 5.37–9.28; *p* < 0.001).

### 3.3. Association between Lean Individuals and LSMM in Patients with NAFLD According to the Sex Differences

In sex-specific subgroup analysis, the presence of LSMM was a significant independent risk factor for lean NAFLD patients, regardless of sex differences. Among the men with NAFLD, the ORs of LSMM for lean NAFLD were consistent after stepwise adjustment for age, DM, hypertension, and WC (Model 1: OR, 10.99; 95% CI, 7.13–17.30; *p* < 0.001); for the presence of IR and CRP (Model 2: OR, 11.12; 95% CI, 7.19–17.55; *p* < 0.001); and for serum alanine aminotransferase, gamma-glutamyl transferase, triglyceride levels, and platelet count (Model 3: OR, 11.12; 95% CI, 7.19–17.55; *p* < 0.001). Among the women with NAFLD, ORs of LSMM for lean NAFLD were maintained after stepwise adjustment for age, DM, hypertension, and WC (Model 1: OR, 4.9; 95% CI, 3.53–6.91; *p* < 0.001); for presence of IR and CRP (Model 2: OR, 4.96; 95% CI, 3.57–7.01; *p* < 0.001); and for serum alanine aminotransferase, gamma-glutamyl transferase, triglyceride levels, and platelet count (Model 3: OR, 5.24; 95% CI, 3.78–7.57; *p* < 0.001) (Table 4).

## 4. Discussion

The present study demonstrated that individuals with lean NAFLD had more favorable metabolic profiles than those with non-lean NAFLD. Stepwise adjustment of traditional risk factors revealed the presence of LSMM as a possible novel factor associated with lean NAFLD subjects, regardless of sex differences.

The prevalence of lean NAFLD is approximately 7–19%, which is associated with different publication dates, definitions of BMI, and variable diagnostic modalities for fatty liver [4]. A recent meta-analysis suggested that the prevalence of non-obese NAFLD was 37.4% (95% CI, 30.6–44.7), and the incidence and prevalence have been gradually increasing over the years [3,21]. In our study on the individuals who underwent health checkups, the prevalence was 41.2%, similar to the previous studies [3,21].

However, the natural history and pathophysiology of lean NAFLD are poorly understood. According to two meta-analyses, NAFLD had metabolic and cardiovascular disease risks, irrespective of lean status, while lean NAFLD had more favorable metabolic risk profiles than non-lean NAFLD [10,22]. In the present study, the lean NAFLD group had lower proportions of comorbidities and more favorable liver and metabolic profiles than the non-lean NAFLD group, showing that our data were consistent with previous results. In addition, multivariate analysis showed that the effects of traditional risk factors, including WC, IR, lipid profiles, and hypertension, in the lean NAFLD group were relatively weak compared to those in the non-lean NAFLD group.

However, several recent reports have suggested that lean NAFLD does not have a better prognosis than non-lean NAFLD [9,23,24]. In a longitudinal study, Hagstrom et al. reported that the lean NAFLD group had a worse prognosis than the non-lean NAFLD group [9]. Despite more favorable metabolic profiles and less severe liver histology in patients with lean NAFLD, the lean NAFLD group had increased severe liver-related events and overall mortality [9]. In other studies, the proportion of non-alcoholic steatohepatitis and significant fibrosis in patients with nonobese NAFLD accounted for approximately 40% and 30%, respectively, which is associated with high mortality in patients with nonobese NAFLD [3,23,24]. Given these previous results, there are unknown factors associated with lean NAFLD that distinguish it from non-lean NAFLD.

We demonstrated that the proportion of LSMM was higher in lean NAFLD patients than in non-lean NAFLD patients. In addition, LSMM was more strongly associated with lean NAFLD than non-lean NAFLD, suggesting its potential as a future risk factor. Previous studies have shown that sarcopenia is associated with NAFLD [13,14]. The putative mechanisms are as follows: Skeletal muscle is an insulin-responsive target organ and a primary site for glucose disposal [25]. LSMM can exacerbate IR and increase lipolysis in adipose tissue with the free fatty acid influx into the liver, aggravating NAFLD [26]. From a skeletal muscle perspective, persistent IR and chronic inflammation mechanisms may promote proteolysis in a vicious cycle [12]. In a longitudinal study, Kim et al. demonstrated that increases in skeletal muscle mass might positively affect the resolution of existing NAFLD [27]. The results suggest that increasing the skeletal muscle mass is a possible treatment strategy for lean NAFLD as well as NAFLD patients.

In terms of demographics, the prevalence of lean NAFLD is higher in the Asian population than in Caucasians, which may be related to the LSMM in Asian populations with relatively lower muscle mass than Caucasians [4]. Alams et al. demonstrated that the effect of weight reduction on hepatic histological activity and fibrosis was limited in patients with lean NAFLD compared to those with non-lean NAFLD, which may be associated with LSMM in patients with lean NAFLD [28].

To the best of our knowledge, we investigated the association between LSMM and lean NAFLD using a sex-specific analysis. NAFLD is characterized by sexual dysmorphism, which is known to worsen in men and postmenopausal women [29]. Estrogen deficiency due to ovarian senescence is closely associated with NAFLD progression, including massive steatosis and fibrotic evolution [29]. In our study, the average age of the women was 47.2 years (data not shown), which was younger than the Korean menopause age of 49.3 years [30]. However, a questionnaire survey on the definite age of menopause is insufficient. Considering that the OR for LSMM in patients with lean NAFLD was relatively lower in women than in men, additional research is required to include menopausal status and estrogen levels.

The limitations of this study are as follows: First, because this was a single-institution, retrospective, and cross-sectional study, the causal relationship between LSMM and lean NAFLD is uncertain, and it is difficult to apply our results to the general population. Second, the potential for selection bias exists because the health promotion center Asian cohorts are concerned about their health and willing to pay for medical care. Third, our study revealed an association between lean NAFLD and LSMM but did not establish a relationship between LSMM and prognosis in lean NAFLD patients. Fourth, although PNPLA3 rs738409 was found to be closely related to the development and progression of non-obese NAFLD as a risk factor for lean NAFLD, genetic factors were not evaluated in this study [31]. Feldmann et al. demonstrated that low levels of IL-6 and high levels of adiponectin are related to metabolic favorites in patients with lean NAFLD [32]. We were unable to identify these factors in the present study. Finally, we could not reveal the effects of LSMM in the obese (non-lean) and non-obese (lean) subgroups as well as in the NAFLD and non-NAFLD groups with or without NAFLD in the total population. Further well-designed studies are required.

## 5. Conclusions

In conclusion, patients with lean NAFLD had greater metabolic superiority than those with non-lean NAFLD. LSMM is more strongly associated with lean NAFLD than non-lean NAFLD, independent of traditional metabolic factors. In addition, LSMM is closely associated with lean NAFLD, regardless of sex. A well-structured, prospective study is needed to establish the relationship between LSMM and prognosis in patients with lean NAFLD and to establish a relationship with lean NAFLD according to increased skeletal muscle mass to suggest a potential new treatment strategy.

## Figures and Tables

**Figure 1 healthcare-10-00850-f001:**
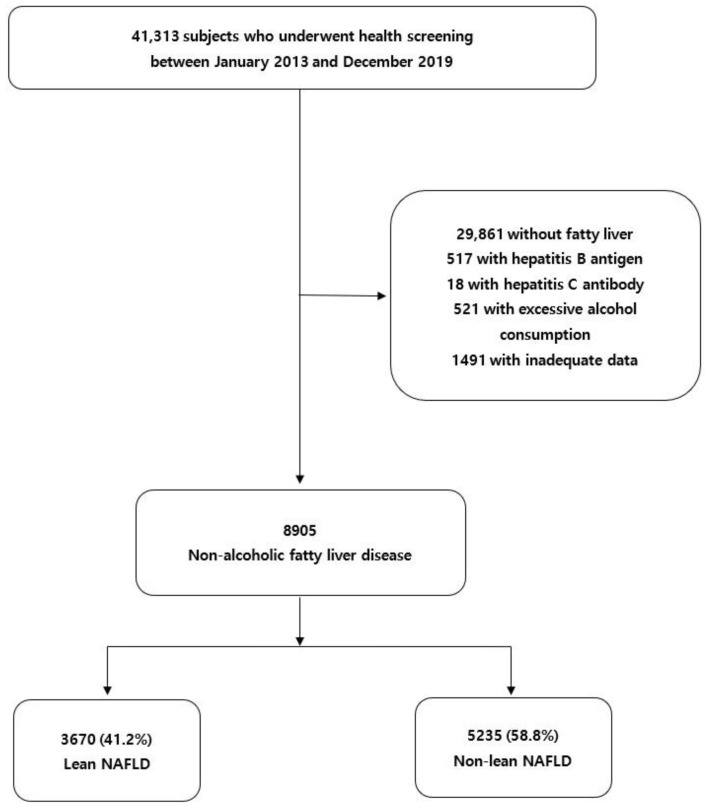
Flow chart of the participants. NAFLD, non-alcoholic fatty liver disease.

**Figure 2 healthcare-10-00850-f002:**
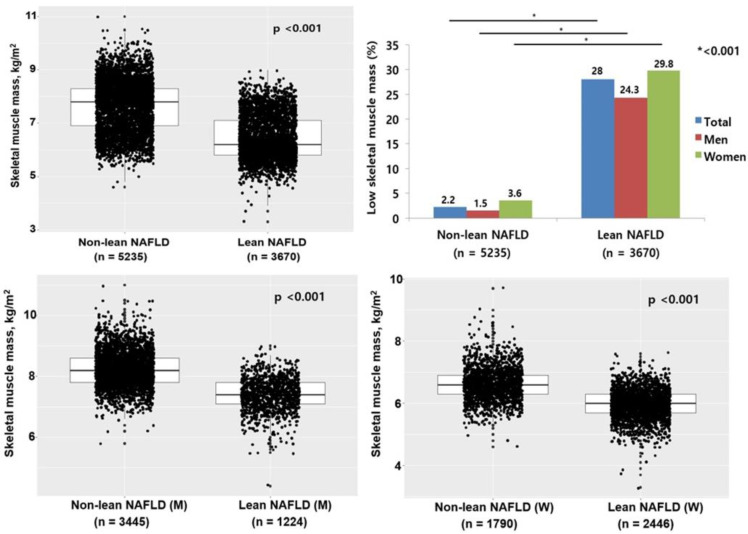
Skeletal muscle mass distribution and percentage of low skeletal muscle mass according to presence and absence of lean NAFLD and sex differences. NAFLD, non-alcoholic fatty liver disease; M, men; W, women.

**Table 1 healthcare-10-00850-t001:** Enrolled patients according to the presence or absence of lean NAFLD.

	Lean NAFLD *n* = 3670 (41.2%)	Non-Lean NAFLD *n* = 5235 (58.8%)	*p*-Value
Anthropometric profiles			
Age, year	45.8 ± 11.6	49.0 ± 11.3	<0.001
Men, *n* (%)	1224 (33.4)	3445 (65.8)	<0.001
BMI, kg/m^2^	20.9 ± 1.5	25.9 ± 2.4	<0.001
WC, cm	73.5 ± 5.4	85.8 ± 6.7	<0.001
Comorbidities			
DM	113 (3.1)	389 (7.4)	<0.001
Hypertension	153 (4.2)	795 (15.2)	<0.001
Metabolic syndrome	137 (3.7)	904 (17.3)	<0.001
Liver profiles			
ALT, IU/L	18.8 ± 11.4	31.9 ± 23.4	<0.001
Platelet count, K/µL	244.7 ± 56.8	247.6 ± 56.4	0.020
GGT, IU/L	21.5 ± 20.2	36.4 ± 36.6	<0.001
Albumin, g/dL	4.7 ± 0.3	4.8 ± 0.3	0.003
Metabolic profiles			
FPG, mg/dL	94.1 ± 17.9	100.0 ± 21.5	<0.001
TG, mg/dL	103.8 ± 57.2	153.1 ± 94.2	<0.001
HDL-C, mg/dL	64.1 ± 14.9	53.5 ± 13.2	<0.001
LDL-C, mg/dL	113.0 ± 33.4	122.7 ± 36.3	<0.001
CRP, mg/dL	0.09 ± 0.34	0.13 ± 0.21	<0.001
HOMA-IR	1.3 ± 0.8	2.0 ± 1.8	<0.001
Body composition profiles			
Skeletal muscle index, kg/m^2^	6.2 ± 0.8	7.8 ± 0.9	<0.001
LSMM, %	1027 (28.0)	115 (2.2)	<0.001

Variables are presented as mean ± standard deviation or numbers (%). NAFLD, non-alcoholic fatty liver disease; BMI, body mass index; WC, waist circumference; DM, diabetes mellitus; ALT, alanine aminotransferase; GGT, gamma-glutamyl transferase; FPG, fasting plasma glucose; TG, triglyceride; HDL-C, high-density lipoprotein cholesterol; LDL-C, low-density lipoprotein cholesterol; CRP, C-reactive protein; HOMA-IR, homeostasis model of insulin resistance; LSMM, low skeletal muscle mass.

**Table 2 healthcare-10-00850-t002:** Risk factors associated with lean NAFLD patients.

Variable	Univariate	Multivariate	
	*p*-Value	*p*-Value	OR (95% CI)
Age, years	<0.001	<0.001	0.99 (0.98–0.99)
Male, yes/no	<0.001	<0.001	0.51 (0.44–0.60)
BMI	0.723		
WC, cm	<0.001	<0.001	0.67 (0.66–0.68)
DM, yes/no	<0.001		
Hypertension, yes/no	<0.001	<0.001	0.52 (0.40–0.68)
Presence of IR	<0.001	0.003	0.68 (0.53–0.87)
ALT	<0.001		
GGT	<0.001		
FPG	<0.001		
TG	<0.001	0.005	0.86 (0.77–0.95)
Platelet count	0.020	0.001	0.82 (0.72–0.93)
CRP	<0.001		
LSMM, yes/no	<0.001	<0.001	6.88 (5.27–9.07)

NAFLD, non-alcoholic fatty liver disease; OR, odds ratio; CI, confidence interval; WC, waist circumference; DM, diabetes mellitus; IR, insulin resistance; ALT, alanine aminotransferase; GGT, gamma-glutamyl transferase; FPG, fasting plasma glucose; TG, triglyceride; CRP, C-reactive protein; LSMM, low skeletal muscle mass.

**Table 3 healthcare-10-00850-t003:** Adjusted OR of low skeletal muscle mass for lean NAFLD patients.

	Low Skeletal Muscle Mass	
	OR (95% CI)	*p*-Value
Total NAFLD population (*n* = 8905), OR for lean NAFLD		
Unadjusted	17.30 (14.25–21.20)	<0.001
Multivariate model 1	18.86 (15.34–23.40)	<0.001
Multivariate model 2	6.79 (5.20–8.97)	<0.001
Multivariate model 3	6.88 (5.26–9.09)	<0.001
Multivariate model 4	7.02 (5.37–9.28)	<0.001

OR, odds ratio; CI, confidence interval; NAFLD, non-alcoholic fatty liver disease; BMI, body mass index. Model 1 was adjusted for age, sex, and BMI. Model 2 was adjusted for diabetes, hypertension, and waist circumference, inclusive of Model 1. Model 3 was adjusted for the presence of insulin resistance and C-reactive protein, inclusive of Model 2. Model 4 was adjusted for serum alanine aminotransferase, gamma-glutamyl transferase, triglyceride levels, and platelet counts, inclusive of Model 3.

**Table 4 healthcare-10-00850-t004:** Adjusted OR of low skeletal muscle mass for lean NAFLD patients according to sex differences.

	Low Skeletal Muscle Mass	
	OR (95% CI)	*p*-Value
Male patients with NAFLD (*n* = 4669), OR for lean NAFLD		
Unadjusted	21.75 (16.14–29.93)	<0.001
Age and BMI-adjusted model	23.9 (17.61–33.09)	<0.001
Multivariate model 1	10.99 (7.13–17.30)	<0.001
Multivariate model 2	11.12 (7.19–17.55)	<0.001
Multivariate model 3	11.12 (7.19–17.55)	<0.001
Female patients with NAFLD (*n* = 4236), OR for lean NAFLD		
Unadjusted	11.29 (8.75–14.81)	<0.001
Age & BMI adjusted model	12.61 (9.68–16.69)	<0.001
Multivariate model 1	4.9 (3.53–6.91)	<0.001
Multivariate model 2	4.96 (3.57–7.01)	<0.001
Multivariate model 3	5.24 (3.78–7.57)	<0.001

OR, odds ratio; CI, confidence interval; BMI, body mass index; NAFLD, non-alcoholic fatty liver disease. Model 1 was adjusted for age, BMI, diabetes, hypertension, and waist circumference. Model 2 was adjusted for the presence of insulin resistance and C-reactive protein, inclusive of Model 1. Model 3 was adjusted for serum alanine aminotransferase, gamma-glutamyl transferase, triglyceride levels, and platelet counts, inclusive of Model 2.

## Data Availability

The data supporting the findings of this study are also available from the corresponding author (M-K.K.) upon reasonable request.

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
