# Peer review of "Association of Low Skeletal Muscle Mass with the Phenotype of Lean Non-Alcoholic Fatty Liver Disease"

_healthcare, 2022, doi:10.3390/healthcare10050850_

Round 1

Reviewer 1 Report

The manuscript "Effect of low skeletal muscle mass lean individuals in patients with non-alcoholic fatty liver disease" by Byeon et. al., reaffirms the connection between low skeletal muscle mass in non-obese patients with NAFLD. This is particularly prevalent in the Asian population. Authors conclude patients with lean NAFLD had a better metabolic profile than non-lean NAFLD, and low skeletal muscle mass is associated with lean NAFLD than non-lean NAFLD. Although this adds to the existing knowledge, it is not entirely a new finding. The study would have a broad interest in future research on how an increase in skeletal muscle mass can prevent lean NAFLD.

However, the manuscript has the following shortcomings:

  1. Table 1 lists baseline characteristics. The title of the table is confusing. Baseline characteristics comparing what with what.
  2. Table 1 needs subheadings on Metabolic profile,  Liver function tests findings, Histological outcomes (Fibrosis, etc), and most importantly Survival outcomes are missing.
  3. Figure 2 legend needs the number of data points for each of the four graphs  (n=?) and the statistical method.
  4. Clarification on the association of NAFLD with low skeletal muscle mass in obese patients when compared to normal individuals is required.

Author Response

< Reviewer 1 >

The manuscript "Effect of low skeletal muscle mass lean individuals in patients with non-alcoholic fatty liver disease" by Byeon et. al., reaffirms the connection between low skeletal muscle mass in non-obese patients with NAFLD. This is particularly prevalent in the Asian population. Authors conclude patients with lean NAFLD had a better metabolic profile than non-lean NAFLD, and low skeletal muscle mass is associated with lean NAFLD than non-lean NAFLD. Although this adds to the existing knowledge, it is not entirely a new finding. The study would have a broad interest in future research on how an increase in skeletal muscle mass can prevent lean NAFLD.

However, the manuscript has the following shortcomings:

  1. Table 1 lists baseline characteristics. The title of the table is confusing. Baseline characteristics comparing what with what. 2. Table 1 needs subheadings on Metabolic profile, Liver function tests findings, Histological outcomes (Fibrosis, etc), and most importantly Survival outcomes are missing.

Response 1-2)

Thank you for careful reading of our manuscript. We appreciate the time and effort that you have dedicated to providing valuable comments on our manuscript. I fully agree with your opinions, and additional comments has been inserted.

Fully agreeing with the vagueness of the expression of baseline characteristics, we revised the title to Enrolled patients according to the presence or absence of lean NAFLD. In addition, to clearly indicate the subheading, what you mentioned including liver and metabolic profiles is expressed in bold (TABLE 1).

However, histologic outcome and survival outcome were limited in description. This was a retrospective, cross-sectional study conducted on a health promotion center-based population, and there were no fatalities and there were disadvantages in that a biopsy could not be performed.  

  1. Figure 2 legend needs the number of data points for each of the four graphs (n=?) and the statistical method.

Response 3)

Thank you for your opinions. As mentioned, the figure was revised by entering new data points, and each used statistical technique was inserted into the text. We sincerely appreciate your comments to help us organize our rich content (Figure 2. & page 4-5, line 140-148).

  1. Clarification on the association of NAFLD with low skeletal muscle mass in obese patients when compared to normal individuals is required.

Response 4)

Thank you for your sharp valuable comments. I fully agree with your opinions.

I will briefly explain the background of the progress of this study.

As is well known, in the Asian population, the patients with lean NAFLD have favorable metabolic components compared to those with non-lean NAFLD, but no significant risk factors are presented except for genetic factors including PNPLA3. This study started from the question of the relationship between the presence of sarcopenia and BMI in patients with NAFLD. Although not described, a low BMI (OR: 0.28, 95% CI, 0.26-0.30, p<0.001) was a risk factor for sarcopenia in our NAFLD cohort. Based on this, the BMI was divided into a lean group and a non-lean group, and the OR of sarcopenia was observed in each group.

It is a limitation that the effect of sarcopenia was not revealed in the obese (non-lean) and non-obese (lean) groups as well as in the NAFLD and non-NAFLD groups in the total population including normal participants, and this was described in the limitation. (Page 10, line 288-291)

I appreciate for your kind and valuable comments. We hope that our revision will meet with approval. We would like to respond to any further questions and comments you may have.

Reviewer 2 Report

The retrospective study carried out by Byeon JH et al. demonstrates that individuals with lean NAFLD had more favorable metabolic profiles than those with non-lean NAFLD. Moreover, they show that low skeletal muscle mass (LSMM) was a risk factor for lean NAFLD, compared with non-lean NAFLD.

Major comments

1.  These results are quite natural because BMI is well known to be associated both with metabolic profiles and with skeletal muscle mass. Therefore, the present form of manuscript may have little value to be published. The authors should analyze the data including BMI. Is LSMM a risk factor independent of BMI?

2.  To elucidate whether LSMM is a risk factor for NAFLD or not, it would be better to investigate the association between skeletal muscle mass and the prevalence of NAFLD using not only data of individuals with NAFLD, but also those without NAFLD.

Minor comments

3.  Please clarify what kind of population this study included. Additionally, the name of the health promotion center should be specified.

Author Response

< Reviewer 2 >

The retrospective study carried out by Byeon JH et al. demonstrates that individuals with lean NAFLD had more favorable metabolic profiles than those with non-lean NAFLD. Moreover, they show that low skeletal muscle mass (LSMM) was a risk factor for lean NAFLD, compared with non-lean NAFLD.

Major comments

  1. These results are quite natural because BMI is well known to be associated both with metabolic profiles and with skeletal muscle mass. Therefore, the present form of manuscript may have little value to be published. The authors should analyze the data including BMI. Is LSMM a risk factor independent of BMI?

Response 1)

Thank you for careful reading of our manuscript. We appreciate the time and effort that you have dedicated to providing valuable comments on our manuscript. I fully agree with your opinions, and additional comments has been inserted.

As you mentioned, we rerun the statistics including BMI, and LSMM was an associated factor for lean NAFLD even after adjusting for BMI. (Table 2,3,4), page 6-8) Thanks for pointing it out.

  1. To elucidate whether LSMM is a risk factor for NAFLD or not, it would be better to investigate the association between skeletal muscle mass and the prevalence of NAFLD using not only data of individuals with NAFLD, but also those without NAFLD.

Response 2)

Thank you for your sharp valuable comments. I fully agree with your opinions.

I will briefly explain the background of the progress of this study.

As is well known, in the Asian population, the patients with lean NAFLD have favorable metabolic components compared to those with non-lean NAFLD, but no significant risk factors are presented except for genetic factors including PNPLA3. This study started from the question of the relationship between the presence of sarcopenia and BMI in patients with NAFLD. Although not described, a low BMI (OR: 0.28, 95% CI, 0.26-0.30, p<0.001) was a risk factor for sarcopenia in our NAFLD cohort. Based on this, the BMI was divided into a lean group and a non-lean group, and the OR of sarcopenia was observed in each group.

It is a limitation that the effect of sarcopenia was not revealed in the obese (non-lean) and non-obese (lean) groups as well as in the NAFLD and non-NAFLD groups in the total population including normal participants, and this was described in the limitation. (Page 9, line 281-295)

Minor comments

  1. Please clarify what kind of population this study included. Additionally, the name of the health promotion center should be specified.

Response 3)

It is specified that this study was conducted in Yeungnam University Hospital as a study targeting Koreans. As the subjects were patients who were interested in health while having the ability to pay for medical care, selection bias was noted in the limitation. (page 2, line 52-54)

I appreciate for your kind and valuable comments. We hope that our revision will meet with approval. We would like to respond to any further questions and comments you may have.

Round 2

Reviewer 2 Report

The Authors have revised the manuscript appropriately according to this Reviewer's comments. I have no further comment.